# P38 Mediates Tumor Suppression through Reduced Autophagy and Actin Cytoskeleton Changes in NRAS-Mutant Melanoma

**DOI:** 10.3390/cancers15030877

**Published:** 2023-01-31

**Authors:** Ishani Banik, Adhideb Ghosh, Erin Beebe, Blaž Burja, Mojca Frank Bertoncelj, Christopher M. Dooley, Enni Markkanen, Reinhard Dummer, Elisabeth M. Busch-Nentwich, Mitchell P. Levesque

**Affiliations:** 1Department of Dermatology, University of Zurich Hospital, University of Zurich, 8091 Zurich, Switzerland; 2Department of Dermatology, University of San Francisco California, San Francisco, CA 94117, USA; 3Functional Genomics Center Zurich, ETH/University of Zurich, 8057 Zurich, Switzerland; 4Institute of Veterinary Pharmacology and Toxicology, Vetsuisse Faculty, University of Zurich, 8057 Zurich, Switzerland; 5Center of Experimental Rheumatology, Department of Rheumatology, University Hospital Zurich, University of Zurich, 8091 Zurich, Switzerland; 6Team Integrative Biology of Immune-Mediated Inflammatory Diseases, BioMed X Institute, 69120 Heidelberg, Germany; 7Max Plank Institute for Heart and Lung Research, 61231 Bad Nauheim, Germany; 8School of Biological and Behavioural Sciences, Queen Mary University of London, London E1 4NS, UK

**Keywords:** melanoma, NRAS mutation, p38 tumor suppressor, mTOR, autophagy, anisomycin

## Abstract

**Simple Summary:**

Mutations in NRAS are the second most common driver mutation in melanoma and lack therapy options. We discovered that p38 plays the role of a tumor suppressor in NRAS mutant melanoma. In this study, we characterized the significance of short and long term p38 activation which can in turn modulate other pathways downstream. An important of effect of long term p38 activation is the phosphorylation of mTOR. As a result of mTOR phosphorylation, a suppression in autophagy and actin remodeling follows. Our candidate tumor suppressor p38 as well as its downstream targets such as phosphorylation of mTOR, actin remodeling and autophagy can be modulated with pharmacologically available small molecules such as anisomycin, rapamycin, hydroxychloroquine and cytochalasin. Further investigation to modulate a combination of these compounds along with FDA approved drugs like MEK inhibitors can be a novel strategy to treat NRAS mutant melanomas.

**Abstract:**

Hotspot mutations in the NRAS gene are causative genetic events associated with the development of melanoma. Currently, there are no FDA-approved drugs directly targeting NRAS mutations. Previously, we showed that p38 acts as a tumor suppressor in vitro and in vivo with respect to NRAS-mutant melanoma. We observed that because of p38 activation through treatment with the protein synthesis inhibitor, anisomycin leads to a transient upregulation of several targets of the cAMP pathway, representing a stressed cancer cell state that is often observed by therapeutic doses of MAPK inhibitors in melanoma patients. Meanwhile, genetically induced p38 or its stable transduction leads to a distinct cellular transcriptional state. Contrary to previous work showing an association of invasiveness with high p38 levels in BRAF-mutated melanoma, there was no correlation of p38 expression with NRAS-mutant melanoma invasion, highlighting the difference in BRAF and NRAS-driven melanomas. Although the role of p38 has been reported to be that of both tumor suppressor and oncogene, we show here that p38 specifically plays the role of a tumor suppressor in NRAS-mutant melanoma. Both the transient and stable activation of p38 elicits phosphorylation of mTOR, reported to be a master switch in regulating autophagy. Indeed, we observed a correlation between elevated levels of phosphorylated mTOR and a reduction in LC3 conversion (LCII/LCI), indicative of suppressed autophagy. Furthermore, a reduction in actin intensity in p38–high cells strongly suggests a role of mTOR in regulating actin and a remodeling in the NRAS-mutant melanoma cells. Therefore, p38 plays a tumor suppressive role in NRAS-mutant melanomas at least partially through the mechanism of mTOR upregulation, suppressed autophagy, and reduced actin polymerization. One or more combinations of MEK inhibitors with either anisomycin, rapamycin, chloroquine/bafilomycin, and cytochalasin modulate p38 activation, mTOR phosphorylation, autophagy, and actin polymerization, respectively, and they may provide an alternate route to targeting NRAS-mutant melanoma.

## 1. Introduction

There is increasing evidence for the rising incidence of melanoma worldwide [1,2]. BRAF is the most commonly mutated gene involved in the development of melanomas, but other RAS superfamily members (HRAS, NRAS, and KRAS) also contribute to the initiation, progression, aggressiveness, and response to the therapy of cutaneous melanoma [3]. For example, 28% of all melanomas have NRAS mutations [4]. Currently, there are no effective targeted therapies for NRAS mutations, although previous strategies have been focused on KRAS [5]. Given that 15–20% of activating NRAS mutations [6] are found among melanoma patients and lack therapy options outside of checkpoint inhibition, our investigations were focused on NRAS-mutant melanomas. NRAS mutations are mostly associated with more aggressive nodular melanoma, similar to our previously described NRAS model in transgenic zebrafish *Tg*(*mitfa*;*NRAS^Q61K^*)*mitfa^w2^*;*tp53^zdf1^* [7,8]. Using our transgenic zebrafish model as a platform for the identification of novel therapeutic targets, we discovered the role of p38 as a tumor suppressor in zebrafish as well as in patient-derived NRAS-mutant melanoma cultures [9]. We reported that p38 overexpression leads to JNK activation, resulting in increased apoptosis. Our observations of reversing MEK inhibitor resistance using low-dose anisomycin in NRAS-mutant melanoma cells led us to investigate the tumor suppressing activity of p38.

P38 is a multi-functional kinase that is activated by a wide range of stimuli [10] and regulates a plethora of physiological and cellular functions, including differentiation, migration, and apoptosis, amongst other roles [11,12]. One of the earliest studies of p38 described it as a tumor suppressor based on its inhibitory activity on RAS-dependent kinases [13], but several reports have described its pro-tumorigenic function [14]. Therefore, the dual role of p38 in tumorigenesis warrants a further elucidation of the critical aspects of p38 biology that may have potential therapeutic implications [15]. Recent studies have demonstrated the role of p38 as a tumor suppressor in thyroid cancer [16], breast cancer [17], the prevention of bone metastasis in prostate cancer [18], and suppressed mammary tumorigenesis [19]. Because p38 regulates more than 60 proteins [20], delineating the downstream targets and effect of p38 activation on MAPK signaling in NRAS-mutant melanoma cells is imperative to better target these cells. Therefore, we tested the expression of several proteins in the JNK, ERK/MEK, and mTOR/AKT pathways, resulting in an overview of transiently and consistently expressed proteins upon p38 activation. The physiological relevance of our investigations lies in the identification of one or more proteins affecting the p38 pathway as well as cellular functions such as autophagy that serve as new therapeutic targets.

## 2. Results

### 2.1. Short- and Long-Term p38 Activation by Anisomycin and Stable Transduction Leads to a Distinct Transcriptional Signature in Patient-Derived NRAS Melanoma Cell Lines

We previously demonstrated the tumor suppressive effects of p38 by stable transfection and pharmacological activation. We showed that p38 activation reduced the in vitro viability of NRAS-mutant melanoma cell lines [9]. To further characterize the transcriptional effects, we performed bulk RNA-seq analyses on two patient-derived NRAS-mutant melanoma cell lines (130429 and 160915). They were stably transfected to induce p38 overexpression (p38–130429/160915) or empty-vector control (EV control–130429/160915) [9]. In parallel to the stable overexpression of p38, 130429 and 160915 were treated with anisomycin; this is a common pharmacological strategy to stimulate p38 activation. The p38–high cells (anisomycin-treated or p38–overexpressed) cells separately cluster from the non-p38–high cells (DSMO or EV control), as observed in the PCA plot. We obtained 2621 and 548 DEGs (differentially expressed genes) when we compared p38–130429/p38–160915 with the EV control–130429/160915, respectively (two-fold change, adjusted *p*-value < 0.05). We also compared anisomycin-treated 130429 and 160915 cells with the DMSO treated cells and obtained 573 and 1527 DEGs, respectively. The short-term (30 min) anisomycin treatment stimulation resulted in well-characterized stress-induced response genes [21], exemplified by the transcriptional upregulation of the downstream targets of the stress-responsive cAMP pathway [22], specifically cJUN, FOS, and ATF3 (two-fold change, adjusted *p*-value < 0.05) (Figure 1A) (full list of DEG pathways by ORA in Appendix A). Furthermore, apoptosis-associated NR4A1 was one of the most strongly upregulated genes upon pharmacological activation of p38. Among the most significantly upregulated pathways were the skeletal muscle differentiation (GO:0035914) and G protein receptor signaling (GO:007186), whereas the cell population proliferation pathway (GO:008283/84) was downregulated (*p*-value < 0.05). Contrary to this stress-induced transient response phenomenon, p38 stably transfected 130429/160915 cell lines, and they were characterized by the upregulation of the extracellular matrix reorganization and cell adhesion pathways (*p* < 0.05) (full list of DEG pathways by ORA in Appendix A). Genes contributing to the increase in the signaling of these pathways include VCAM, NCAM, MMPs, and COL5A1 (Figure 1B). Among the most significantly upregulated pathways were the extracellular matrix reorganization (GO:0030198) and cell adhesion (GO:0007155) pathways, whereas the drug metabolism pathway (hsa00983) was downregulated (*p*-value < 0.05). In conclusion, our RNA sequencing results indicated that a stable p38 upregulation might yield a transiently activated signaling cascade or a lasting activation of genes involved in matrix reorganization and cell adhesion depending on the type or timing of p38 activation. Stimulating the p38 pathway by short-term treatment with anisomycin is a common strategy to study the role of p38. Often after long-term treatment (24 h or more) with anisomycin, the effect of p38 activation is diminished. Here, we showed that there are significant differences in the pathways associated with p38 signaling depending on the method of activation. Taken together, our RNA sequencing results indicated that p38 perturbations produce a change in the extracellular matrix.

### 2.2. Activation of p38 in 160915 Spheroids Does Not Increase Invasion

Previous studies have reported increased invasion of cancer cells with p38 overexpression [23,24,25]. We have previously shown that high levels of p38 in NRAS-mutant melanoma cells reduces proliferation and induces apoptosis while activating the JNK pathway, contrary to the study reported by Puujjalka et al. [26]. Moreover, there have been reports of increased invasion in melanoma cells due to increased p38/MK2 activity [27], but this could be strongly linked to the use of BRAF-mutant cells while investigating the effects of p38 [28]. We cultured empty vector (EV) control and stably transfected p38 cell lines as spheroids and examined their invasiveness on a layer of collagen.

We observed that the EV control–160915 spheroids were considerably larger than the p38–160915 spheroids at 24, 48, and 72 hours post-embedding in collagen (Figure 2A). The viability of p38–160915 spheroids was less than the EV control–160915 spheroids, and the viability of the EV control–160915 spheroids reduced with anisomycin treatments (Appendix A). The total area of the spheroids in the EV control–160915 was significantly larger than p38–160915 with and without anisomycin treatment (Figure 2B). Next, we assessed the difference in fragmentation and observed that the EV control–160915 spheroids had an increase in fragmentation with time, which receded when treated with anisomycin. The same trend was observed in p38–160915 spheroids, although their fragmentation change was marginal compared with that of EV control–160915 prior to anisomycin treatment (Figure 2C). A similar trend was observed when we calculated the invasion distance, where EV control–160915 spheroids have a significantly higher invasion distance compared with p38–160915 spheroids under anisomycin treatment (Figure 2D). Invasion distance was calculated as the distance of single cells invading from the core of the sphere to the periphery or into the collagen. When we compared the EV control–160915 group alone under DMSO and anisomycin treatment conditions, we noticed that although there was no difference in the total area of the spheroids, the fragmentation and invasion distance were remarkably decreased under anisomycin treatment (Figure 2E–G). From the collagen invasion assay of the spheroids, we concluded that p38 does not increase invasion, but rather decreases invasion in NRAS-mutant melanoma cells.

As lower levels of the melanoma specific transcription factor MITF (micropthalmia-associated transcription factor) are associated with increased invasiveness in melanoma cells [29,30], we examined the protein levels of MITF in 130429 and 160915 cell lines 30 min and 24 h post-0.1 µM anisomycin treatment; however, we did not observe any difference (Figure 3, top and middle panel). For all immunoblotting experiments, we treated the p38–160915/130429 cells for 30 min as longer incubation times with anisomycin resulted in unviable cells with limited amount of cell lysate. We observed that p38–130429 cells appear to have lower levels of MITF when treated with anisomycin for 30 min (Figure 3, second from bottom panel), but there was no overall difference in the MITF levels between the EV-160915 and p38–160915 cells (Figure 3, bottom panel). Whether the lower MITF levels would lead to an increased invasion of spheroids remains unknown as the 130429 cells could not be transformed into spheroids.

### 2.3. Activation of p38 Does Not Affect Beta-Catenin Localization or Cell Motility

To further investigate the negative role of p38 in NRAS-mutant melanoma invasion, we performed two assays: immunofluorescence staining of beta-catenin and a motility assay. The Wnt5/β-catenin signaling pathway is a critical determinant of invasive properties in melanoma cells as has been summarized by Webster et al. [31]. To assess the amount of beta-catenin or a change in its localization within the cell, we performed an immunofluorescence staining to visualize the beta-catenin with and without anisomycin/DMSO treatments in different cell lines. As expected, beta-catenin was localized in the nucleus of the cells. There was no change in size, shape, amount, or localization of beta-catenin under anisomycin treatment conditions at 30 min or 24 h in both 130429 and 160915 cell lines (Figure 4). Additionally, we did not observe any changes in the beta-catenin levels or localization in EV control–130429/160915 in comparison to p38–130429/160915 (Figure 4). Thus, p38 levels do not influence beta-catenin in NRAS-mutant melanoma cells and beta-catenin mostly remains localized in the nucleus.

We then performed a motility assay to evaluate if p38 levels influenced cell migration (Figure 5A). There was no significant difference within the cell counts of treated and untreated 130429/160915 cells and within the EV control versus p38–130429/160915 (Figure 5B,C), except for between the anisomycin-treated EV control–130429 and those treated with DMSO. As we wanted to rule out effects specific to cell line groups (i.e., 130429 and 160915), we performed the assay on a different patient-derived NRAS-mutant melanoma cell line 130227. Consistent with our previous results, we did not find a significant difference in the cell counts under untreated, DMSO-treated, and anisomycin treatment conditions (Figure 5D). Taken together, these results point to no effect of p38 levels on migration and invasion of NRAS-mutant melanoma cells. Collectively, we demonstrated the invasion tendency of NRAS-mutant melanoma cells in 2D assays and 3D spheroids, including the assessment of bona fide markers of invasion in melanoma cells, and found that p38 does not increase invasion in NRAS-mutant melanoma cells.

### 2.4. p38 Activation Results in Transient Activation of Phospho-p90 and AKT and Stable Activation of Phospho-mTOR

P38 is a multifunctional kinase and can simultaneously activate several pathways. To understand the physiological relevance of p38 activation in the context of NRAS-mutant melanoma, we interrogated a list of active pathways and proteins with a Proteome Profiler Human Phospho-Kinase Array Kit (R&D systems), which is a membrane-based sandwich immunoassay capable of detecting 45 phosphorylations in a single sample. We performed the assay on 160915 and p38–160915 under 30 min and 24 h anisomycin treatment conditions (Appendix A). A densitometry analysis revealed significant differences in the protein levels of ERK, cJUN, GSK3, RSK1/2/3, WNK1, and YES (Appendix A). AKT and pRAS40 protein levels appeared to be strongly upregulated under short-term anisomycin treatment. We validated some of the targets of the assay using Western blots (Appendix A) and narrowed the list to mTOR, AKT, p90, and cJUN. We observed a transient increase in the phosphorylation of AKT and p90 when the cells were treated for 30 min with anisomycin. Meanwhile, phosphorylation of mTOR was elevated up to 24 h post-anisomycin treatment in both cell lines. Although the phosphorylation of cJUN was remarkably high in 130429 when treated for 30 min, the same effect was not observed in 160915 (Figure 6A,B). Furthermore, we observed an increase in phospho-AKT, p90, cJUN, and mTOR in both EV control and p38–130429/160915 cells when treated with anisomycin (Figure 6C,D). Interestingly, untreated p38–130429/160915 cells had high basal levels of phospho-mTOR compared with untreated EV control–130429/160915 cells. These results indicate that anisomycin induced a stress response signaling that was demonstrated by the transient protein phosphorylation of cJUN, p90, and AKT. However, p38 upregulation leads to a lasting increase in the phosphorylation of mTOR.

### 2.5. Phosphorylation of mTOR Leads to Reduced Autophagy

On the one hand, activation of mTOR promotes anabolism and protein synthesis by phosphorylating S6K (p70) [32]. On the other hand, it promotes catabolism by the phosphorylation of the ULK-Atg13 complex [33].

From our preliminary analysis of the protein expression of autophagy-associated genes ATG4, BECLIN, p62, and LC3 (Appendix A), we identified differences in the expression of p62 and LC3 with and without anisomycin in our cell lines. The accumulation of p62 is a good indicator of autophagic flux [34,35], while the amount of LC3 (LC3-I to LC3-II conversion) correlates to the number of autophagosomes [36]. Hydroxychloroquine (HCQ) is the only clinically approved autophagy repressor that inhibits autophagy due to lysosomal acidification and subsequently blocks the fusion of autophagosomes that accelerates cell death [37]. High levels of LC3 conversion under HCQ treatment for 24 h in comparison to the untreated conditions indicated functional autophagy in all cell lines (Figure 7A–D). Consistent with the transient nature of p38 stress response signaling, LC3 conversion was transiently reduced when the cells were treated with anisomycin for 30 min, indicating impaired degradation or increased autophagosome formation (Figure 7A,B). By the 24 h anisomycin treatment time point, when p38 hyperactivity is nearly lost, reduced LC3 conversion begins to recover. The LC3 conversion is also reduced when EV control–130429/160915 and p38–160915 cells are treated with anisomycin for 30 min. Interestingly, we did not observe any band in p38–130429 when probed for LC3 and p62, possibly due to very low numbers of autophagosomes because of the consistently high levels of p38, explaining the lack of difference in LC3 conversion with anisomycin treatment. (Figure 7C,D). In summary, our results indicate a transient autophagy deficiency upon anisomycin treatment. Although our assessment of LC3 conversion points in the general direction of reduced autophagy as a result of p38 overexpression, we cannot rule out the fact that impaired autophagy cannot be demonstrated by the protein expression of LC3 alone. Moreover, further investigation would be necessary to conclude if impaired autophagy observed by the reduced LC3 conversion is due to the very high turnover of LC3 or fewer autophagosomes to begin with.

### 2.6. RNA Sequencing of Transgenic Zebrafish Overexpressing p38 and Actin Staining In Vitro Suggests Actin Remodeling with p38 Overexpression

We had previously generated transgenic zebrafish expressing *MITF*-driven NRAS^Q61*K*^ as well as *MITF*-driven p38 in combination with NRAS^Q61K^ (p38–NRAS^Q61K^) [9]. Micro-dissected FFPE (formalin-fixed paraffin embedded) tumor tissues from *Tg*(*mitfa:p38α*);*Tg*(*mitfa:NRAS^Q61K^*);*mitfa^w2^*;*tp53^zdf1^* (p38–NRAS^Q61K^, in short) versus *Tg*(*mitfa:NRAS^Q61K^*);*mitfa^w2^*;*tp53^zdf1^* (NRAS^Q61K^, in short) were clustered based on RNA sequencing and observed 517 DEGs (two-fold change, *p*-value < 0.01) (Figure 8A). We observed an upregulation in actin binding genes such as *actn3a*/*b* and *myh* in p38–NRAS^Q61K^. Consistent with the upregulated genes and pathways observed in p38–high cells, we observed elevated levels of *col10*, *fos*, *cxcl*, and *erg1* as well as upregulated extracellular matrix reorganization in p38–NRAS^Q61K^ (Figure 8B). The most upregulated pathways in p38–NRAS^Q61K^ were extracellular matrix reorganization (GO:0030198), actin binding (GO:0051371), and cell adhesion (GO:0007155), which is concordant with our findings of differentially expressed pathways in p38–130429/160915 cell lines (Figure 8C).

Previously, we have reported spindle-shaped nuclei in NRAS^Q61K^ driver tumors enriched in p38 expression in transgenic zebrafish *Tg*(*mitfa:p38α*);*Tg*(*mitfa:NRAS^Q61K^*);*mitfa^w2^*;*tp53^zdf1^*, suggesting the changing dynamics of actin cytoskeleton via p38 activation. Additionally, we observed morphological differences while culturing p38 high cells in vitro. These observations combined with the RNA sequencing results of the extracellular matrix reorganization and actin binding prompted us to explore the differences in actin fiber polymerization upon p38 activation. Following our lead from the RNA sequencing analysis, we observed differences in actin intensity and actin distribution between anisomycin-treated and untreated 130429/160915 cell lines (Appendix A). Under untreated conditions, the cells have a nearly equal distribution of actin in all the cells as well as undisturbed actin fibers connected cell-to-cell (Appendix A, white arrows). With anisomycin treatment, the cell shape starts changing, and they appear to be elongated with uneven cortical and cytoplasmic actin deposits (Appendix A, blue arrows). Furthermore, actin distribution appears polarized and capped (Appendix A, green arrows) [38], causing intracellular actin filaments to be disconnected (Appendix A, yellow arrows). The actin MFI (mean fluorescence intensity) was significantly reduced (*p* < 0.0001) when 130429 cells were treated with anisomycin, while the p38–130429 cells had low MFI in comparison to EV control–130429 under all conditions, although no significant reduction was observed upon anisomycin treatment in p38–130429 (Appendix A, graphs). The actin MFI in anisomycin-treated 160915 cells was also significantly low (*p* = 0.03) and showed a slight reduction in EV control and p38–160915 cells. (Appendix A, graphs). Taken together, in vivo sequencing data coupled with experimental actin staining in vitro suggests that increased p38 levels promotes actin remodeling.

## 3. Discussion

We had previously identified p38 as a druggable tumor suppressive gene in NRAS-mutant melanomas. p38 is a multi-functioning kinase and has been described to have both oncogenic and tumor suppressive roles. The P38 pathway is part of one of the subfamilies associated with stress-induced MAPK signaling. A common strategy to activate p38 is the short-term use of protein synthesis inhibitors such as anisomycin. However, results from p38 activation by short-term protein inhibitor treatments must be interpreted with caution as they are distinct from the long-term overexpression of p38. We investigated the long-term effects of p38 overexpression by treating NRAS-mutant melanoma cells for 24 h with anisomycin, by stable transfection of p38 in cell lines, and by transgenic modification in *Tg*(*mitfa*;*NRAS^Q61K^*)*mitfa^w2^*;*tp53^zdf1^* zebrafish. Melanoma cells have a well-defined stressed cell state, which is marked by the upregulation of immediate early transcription factors (JUN, FOS, ATF3, NR4A1, DUSP) linked to melanoma oncogenic programs, inflammation, and stress response [21]. As expected, we observed a similar stressed cell state in short-term 130429/160915 anisomycin treated cells marked by the upregulation of JUN/JUNB, FOSA/B, ATF3, ERG1/3, NR4A1, and DUSP2/5. Some of these transcription factors are regulated by cAMP signaling and are implicated in BRAF-mutant MEK inhibitor-resistant melanomas [39]. We report here that the long-term effects of p38 by stable transfection have additional lasting effects such as changes in extracellular matrix reorganization and actin remodeling, which were not observed after short-term anisomycin treatment. The RNA sequencing analysis led to the conclusion that distinct pathways are activated depending on the duration of p38 activation and that the stressed cancer cell state observed upon p38 activation is short-lived. Results from using p38 stimulators/inhibitors as well as siRNA-mediated depletion are the basis of several studies, and their discrepancy from the short-term transient induction of p38 signaling may lead to contradictory results.

Wenzina et al. [27] have demonstrated the association of invasive melanoma phenotype with the expression of CDH1 and MK2, both of which are modulated via the p38 pathway. While we found PODXL, a downstream target of MK2, to be significantly upregulated in p38–130429/160915 cells, it was associated with a poor survival of patients [27]. In this study, we have collectively demonstrated four lines of evidence to show the non-involvement of p38 in the invasion of NRAS-mutant melanoma cells. We did not observe an increase in the invasion of stably transfected spheroids overexpressing p38, nor did we observe increased invasion upon pharmacological activation via anisomycin in terms of total area, fragmentation, and invasion distance. A critical role for Wnt5/β-catenin signaling is the activation of MITF, which regulates melanocyte survival, differentiation, and proliferation and most importantly defines EMT (epithelial to mesenchymal transition) states (i.e., proliferative/melanocytic or invasive/mesenchymal) in melanoma cells [40]. Some studies have suggested a pro-migratory and pro-invasive function of beta-catenin in human organotypic skin reconstructs [41] and enhanced invasiveness as a result of increased Wnt signaling [42], while other reports have predicted a poor outcome for patients with the loss of beta-catenin [43,44,45,46]. We did not see a decrease in MITF expression at the protein or RNA levels, which is generally associated with mesenchymal melanoma phenotypes. Moreover, we did not observe any changes in the localization of beta-catenin nor a difference in the motility of p38 high cells. A plausible explanation for our contrary results to the results obtained by Wenzina et al. in terms of invasion might be explained by the fact that we only used NRAS-mutant melanoma cell lines in our study while Wenzina et al. used BRAF^V600E^ cell lines. These observations point to distinct BRAF-driven and NRAS-driven melanoma subtypes that require further investigation. Furthermore, Wenzina et al. used drug-based and siRNA-mediated methods to modulate the p38 levels in cell lines, both of which transiently activate the p38 signaling cascade. Although the function of p38 is specific to cell type and also to the type of stimulus [47], it is for the first time that we report here that the function of p38 can also be gene-specific. For instance, p38 plays a tumor suppressive role in NRAS-mutant melanoma but not in BRAF-mutant melanoma, even though BRAF and NRAS are part of the same MAPK pathway. Supporting this theory, a study reported that p38 activity was required in H-RAS but not N-RAS MCF10F breast epithelial cells [25]. In this context, we were limited in studying the role of p38 in association with only one or two mutations: NRAS mutation in cell lines and *tp53*^−/−^ along with NRAS^Q61K^ in zebrafish. The definitive role of p38 might be better defined by examining it in the presence or absence of additional well-known oncogenes and tumor suppressors in melanomagenesis.

The use of low-dose anisomycin for a short time elicits a transient response in the mTOR/AKT pathway as demonstrated by immunoblotting assays. However, to identify the direct binding partners of p38 in the context of NRAS-mutant melanoma cells, more sophisticated experiments such as mass spectrometry or other protein pull-down assays would be necessary. Although limited by the lack of a detailed phospho-kinome profile of p38’s interaction with its downstream targets, our immunoblotting assay indicates that p90, cJUN, and AKT are transiently phosphorylated while phosphorylation of mTOR lasts up to 24 h post-anisomycin treatment and via stable transfection of p38. We report here that irrespective of the method of p38 activation, the physiological relevance of p38 perturbations results in a lasting change of phospho-mTOR.

mTOR is one of the key regulators of autophagy, and inhibition of mTOR by rapamycin results in the activation of autophagy [48,49,50]. Ganley et al. first reported that Atg13 and ULK1 are phosphorylated by mTOR, and the Atg13-ULK complex acts as a node for integrating incoming signals into autophagosome biogenesis [51]. Autophagy is a catabolic process required to maintain cellular homeostasis by the degradation of long-lived proteins and damaged organelles by lysosomes [36,51]. Autophagy is a key survival feature of cancer cells and is a topic with much interest for programmed cell death in cancer cells [52]. Although we observed a transient autophagy deficiency in p38 high cells, it is hard to interpret the results from immunoblotting assays alone. In this study, we assessed autophagy levels by the number of autophagosomes quantified by LC-I to LC-II conversion; this may not be a bona fide technique to interpret autophagy levels. A recent study found that in the absence of LC3, endogenous LC-I can form a complex with p62 and can accumulate to form inclusion bodies, resulting in increased LC3 positive structures, sometimes confused with autophagosomes. This phenomenon even takes place under partial autophagy compromises such as the short-term knock-down of core autophagy genes [53] or, in our case, with short-term anisomycin treatment. Therefore, our results need to be interpreted carefully, and further investigation is needed in the field of autophagy regulation within the context of mTOR activation by p38. In line with our findings, a recent investigation used a combination of trametinib and chloroquine and observed an abrogation of PDA (pancreatic ductal carcinoma) as well as NRAS-mutant melanoma. In fact, they were able to achieve a striking disease response with this combination in a PDA patient [54]. Overall, our results point in the direction that p38 induces tumor suppressive effects in NRAS-mutant melanoma at least partially through the route of suppressed autophagy.

Numerous studies have shown the association of mTOR signaling to the regulation of the actin cytoskeleton [55,56,57]. The most significant DEGs in our cohort of p38–NRAS^Q61K^ transgenic zebrafish were *actn3a*/*b, myha,* and *myhz*, which were associated to actin binding and extracellular matrix reorganization concordant with our observations of RNA-seq analysis of p38–130429/160915 cell lines. Evidence of p38–regulated actin remodeling via p38–JNK-YAP in lung tissue [58] and via ERK1/2-p38 in articular chondrocytes [59] has been previously reported. Actin staining in anisomycin-treated 130429/160915 and p38–130429/160915 cells incline towards actin depolymerization, although a higher magnification of the microscope and timelapse of the cells would be necessary to make significant quantifiable conclusions. It has been previously reported that the inhibition of actin polymerization and actomyosin tension in melanoma cells suppresses both YAP/TAZ activation and PLX4032 resistance [60]. Therefore, it might be worth investigating actin polymerization regulation using pharmacologically available drugs such as cytochalasin in NRAS-mutant melanomas.

Our data suggest that the mechanism of action of p38–induced tumor suppressive effects in NRAS-mutant melanoma is through the activation of mTOR pathway, which in turn leads to suppressed autophagy and changes the dynamics of actin cytoskeleton. Each of these phenomena, i.e., the activation of p38, mTOR signaling, autophagy, and actin depolymerization, are druggable with compounds such as anisomycin, rapamycin, chloroquine, and cytochalasin. Given that NRAS mutations are difficult to target, melanoma with NRAS mutation might be targeted using one or a combination of these drugs along with FDA-approved MEK inhibitors.

## 4. Methods

### 4.1. Cell Lines and Culture Conditions

The patient-derived melanoma cell lines were provided by Melanoma Biobank, University Hospital Zurich, which were derived according to previously described methods (Raaijmakers et al. 2015). Informed consent was obtained from all patients, and all experiments conformed to the principles set out in the WMA Declaration of Helsinki and Department of Health and Human Services Belmont Report. The use of material for research purposes was approved by the corresponding cantonal ethics commissions (Zürich Biobank): EK-687 and 800. These cell lines are available in the URPP biobank, University of Zurich, Zurich, Switzerland. The expression data are available at http://tcgabrowser.ethz.ch:3839/MCE/ (accessed on 20 November 2022). All melanoma cell lines were cultured in RPMI1640 medium (Gibco, Billings, MT, USA) supplemented with 5% fetal bovine serum (Gibco, Billings, MT, USA), 2 mM of L-glutamine (Biochrome AG, Berlin, Germany), and 50 mg/mL of Normocin (invivoGen, San Diego, CA, USA), hereafter referred as complete melanoma medium. HEK293T cells were cultured in DMEM medium (Gibco, Billings, MT, USA) supplemented with 5% fetal bovine serum and 2 mM of L-glutamine. All cell lines were maintained at 37 °C in a humidified 5% CO2 atmosphere. Anisomycin and Hydroxychloroquine were obtained from Cell Signaling. Stable transfection processes for producing EV control–130429/160915 and p38–130429/160915 have been previously described in (Banik et al. 2020).

### 4.2. Collagen-Invasion Assay in Spheroids

To make spheroids, cells were trypsinized and seeded at a density of 5000 cells/well in a low-attachment, round-bottom 96-well plate (Corning, Somerville, MA, USA). Approximately 24 h post-seeding or until the distinction between individual cells was not possible in the spheroids, they were transferred to a collagen-coated 96-well plate. Collagen solution was prepared on ice using: DMEM, 200 nM of L-glutamine, 10% FBS, 7.5% sodium bicarbonate (Gibco, Billings, MT, USA), 3 mg/mL of collagen (Corning, Somerville, MA, USA, Cat Nr: 354236) and 50 mg/mL of Normocin. We placed 80 µL of collagen solution into one of the 96 wells, where it was allowed to polymerize for 30 min at room temperature. The spheroids were collected with minimum medium using a pipette tip from the low-attachment plate and were inserted on top of the polymerized collagen layer, trying not to make holes. Finally, 50 µL of the medium for melanoma was added in each well. The scaffold was overloaded with the appropriate melanoma medium. Spheroids were monitored for up to 124 h, and pictures were taken and analyzed by using ImageJ (Wayne Rasband) and PhotoshopCS2 (Pantone Inc., Carlastadt, NJ, USA) software. Before imaging the spheres under a confocal microscope, they were stained with calcein (Sigma CatNr: 17783 MW: 994.86) and ethidium homodimer (Sigma Cat Nr: 46043 MW:994.86) with a final concentration of 8 µM and 4 µM, respectively, and the spheres were incubated for 1 h at 37 °C inside the incubator. Calcein stains live cells in fluorescence green while ethidium stains dead cells in fluorescence red.

The spheres were photographed with an optical microscope at different times (24, 48, and 72 h post-embedding in collagen and 24 and 48 h post-treatment). The anisomycin and DMSO (Sigma-Aldrich, St. Louis, MI, USA) treatments were always performed 24 h after seeding the spheroids in collagen. At least 3 spheres were selected with respect to each condition and time point for confocal imaging. The spheres can only be imaged once after being stained with calcein or ethidium, as the dyes invade the entire collagen matrix within 24 h. Images taken by the optical microscope were used for the calculation of the total area and fragmentation, while the data recorded by the confocal microscope were used for the calculation of invasion distance. To calculate the total area, the digital image was processed using Photoshop, brought to a resolution of 300 pixels/inch, and converted to the TIFF file format. Thus, the image was opened with ImageJ software, and it was converted to 8-bit images in order to obtain the image in black and white. Then, the binary images were subjected to a clean-up procedure to eliminate artefacts. Subsequently, the application “threshold” was adjusted, which allowed us to obtain the area of interest in a uniform manner, such as a saturated zone of black, in order to calculate both the total area and the area of the core of the sphere using the “Analyze particle” tool. The area was calculated as the number of total pixels.

The factor shape refers to a value that is affected by an object’s shape but is independent of its dimensions. It provides a value of 1 for a perfect circle and larger values for more invasive spheroids. Fragmentation was calculated as the uniform expansion of the spheroid at 48 and 72 h from its initial state at 24 h post-embedding in collagen. The percentage of fragmentation represents the percent of single or clustered cells released from the total spheroid area or core area. To calculate fragmentation, the formula (invasion area/total area)*100 was used. For each experiment, at least 3 photos/condition were analyzed and at least 3 spheres were embedded. Each experiment was performed in triplicate. Figure 2 is representative of one experiment. Pictures of spheroids were analyzed using ImageJ software (Wayne Rasband) as previously indicated [61].

For the counting of invasive cells, the image was opened with Photoshop and set to a resolution of 500 with +30 contrast. Using the “brush” tool, the edge of the spheroid (the portion that appeared be more compact) was designed, leaving the cells that invade the collagen outside. Subsequently, the command “scale” was used to fix the starting point at the center of the sphere and dragged to the periphery where single cells were seen invading the collagen (if any).

The extent of invasion was assessed by the total area, percentage of fragmentation, and invasion distance, as previously described by Hendrix et al. [62].

The successful embedding of spheres was only possible in the 160915 cell line. The invasion assay was performed under low- and high-anisomycin treatments (0.01 µM and 0.1 µM, respectively) in the case of the EV control, while p38–160915 spheroids could only tolerate a low dose of anisomycin.

### 4.3. Motility Assay

The cells were starved for up to 72 h prior to seeding with starving medium (RPMI1640 without FCS). Next, uncoated 8 µM inserts for 24-well plates were rehydrated in starving medium for 2 h. The cells were trypsinized and seeded in starving medium at a density of 5 × 10^4^ cells/insert. Chemoattractant medium (RPMI1640 with 10% FCS) was added to the bottom of a 24-well plate, and the inserts with cells were placed on top. After 24 h of incubation, the inserts were washed in PBS solution by being placed into another 24-well plate filled with PBS. Non-invading cells were removed from top of the insert using a cotton swab. The inserts were then placed into a 24-well plate containing 500 µL of PBS and 0.1% Triton X (Sigma-Aldrich, St. Louis, MI, USA) + DAPI (1:1000, BD) for 15 min at room temperature, followed by washing in PBS. The membrane from the inserts were removed with a scalpel and mounted on a slide. At least 5 images were recorded per membrane. Experiments were performed in triplicate. The images were converted to 8-bit images in ImageJ software, and the command “cell counter” was applied after adjusting for the background.

### 4.4. RNA Extraction

Cells were grown in a T75 flask until confluency and treated with 0.01 µM of anisomycin for 30 min or 24 h or with DMSO for 30 min before trypsinizing them and collecting them for RNA extraction. Cells were then lysed using 350 µL of lysis buffer provided by the High Pure RNA Extraction Kit (Roche, Basel, Switzerland). The RNA was extracted according to the manufacturer’s instructions, and the RNA was measured using Qbit.

### 4.5. Laser-Capture Microdissection (LCM)

We previously generated transgenic zebrafish expressing only *MITF*-driven NRAS^Q61K^ as well as *MITF*-driven p38 in combination with NRAS^Q61K^ (p38–NRAS^Q61K^) [9], both of which produced tumors. Once tumor growth was visible, the animals were euthanized, and the tumor was excised and placed in 10% formalin. Subsequently, they were embedded in paraffin to form blocks. The LCM technique was used for the selective isolation of tumor and matched non-tumor tissue from the zebrafish FFPE specimen. Regions of interest were reviewed by a veterinary pathologist. The LCM technique was performed using an ArcturusXT Laser Capture Microdissection system with Arcturus CapSure™ Macro LCM caps (ThermoFisher Scientific, Waltham, MA, USA) as previously described [63,64,65,66]. RNA from the LCM-isolated tissue was extracted using the Covaris^®^ truXTRAC™ FFPE microTUBE RNA kit with ultrasonication using Covaris^®^ E220, as previously described [64]. RNA quality was analyzed using high sensitivity RNA ScreenTape^®^ with the Agilent 4200 TapeStation system.

### 4.6. Library Preparation

The quality of the isolated RNA from cell lines was determined with a fragment analyzer (Agilent, Santa Clara, CA, USA). Only the samples with a 260 nm/280 nm ratio between 1.8 and 2.1 and a 28S/18S ratio between 1.5 and 2 were further processed. The TruSeq Stranded mRNA (Illumina, Inc., San Diego, CA, USA) was used in the succeeding steps. Briefly, the total RNA samples (100–1000 ng) were poly A-enriched and then reverse-transcribed into double-stranded cDNA. The cDNA samples were fragmented, end-repaired, and adenylated before the ligation of TruSeq adapters containing unique dual indices (UDI) for multiplexing. Fragments containing TruSeq adapters on both ends were selectively enriched with PCR. The quality and quantity of the enriched libraries were validated using the fragment analyzer (Agilent, Santa Clara, CA, USA). The product was a smear with an average fragment size of approximately 260 bp (base pair). The libraries were normalized to 10 nM in 10 mM of Tris-Cl, pH 8.5, with 0.1% Tween 20. Methods for the library preparation of microdissected FFPE samples were followed as previously described [63].

### 4.7. RNA Seq Analysis

RNA sequencing was carried out at the Functional Genomics Center Zurich (FGCZ) on an Illumina NovaSeq 6000 instrument. The raw sequencing reads were trimmed off of adapter sequences and low-quality bases for quality control using the fastp v0.20 tool (S. Chen et al. 2018) with the parameters --thread 8 --trim_front1 0 --trim_tail1 0 --average_qual 0 --adapter_fasta adapters.fa --max_len1 0 --max_len2 0 --disable_trim_poly_g --trim_poly_x --poly_x_min_len 10 --length_required 18 --compression 4. The high-quality human samples were mapped against the human reference genome assembly GRCh38 using the STAR v2.7.a tool (Bray et al. 2016) with the parameters --sjdbOverhang 150 --outFilterType BySJout --outFilterMatchNmin 30 --outFilterMismatchNmax 10 --outFilterMismatchNoverLmax 0.05 --outMultimapperOrder Random --alignSJDBoverhangMin 1 --alignSJoverhangMin 8 --alignIntronMax 100000 --alignMatesGapMax 100000 --outFilterMultimapNmax 50 --chimSegmentMin 15 --chimJunctionOverhangMin 15 --chimScoreMin 15 --chimScoreSeparation 10 --outSAMstrandField intronMotif --alignEndsProtrude 3 ConcordantPair. Gene expression values were quantified using the R (v4.0.3) package Rsubread v2.4.2 (Dobin et al. 2013). The pseudo-alignment of high-quality zebrafish samples against the zebrafish reference genome assembly GRCz11, and the quantification of gene level expression was performed using Kallisto v0.46 [67] with the parameters -t 8 --bias --bootstrap-samples 10 --seed 42 --single --rf-stranded --fragment-length 180 --sd 50. The R (v4.0.3) package edgeR v3.32.1 [68] was used for a differential gene expression analysis based on a negative binomial generalized linear model approach for all samples. Genes showing at least a 2-fold change difference with significant adjusted (Bonferroni correction) *p*-values were considered to be significant (*p* < 0.05 for human; *p* < 0.01 for zebrafish). A functional enrichment analysis for gene ontology and KEGG pathway terms was performed using R (v4.0.3) package clusterProfiler v3.18.1 [69]. 

### 4.8. Phospho Kinase Assay

We trypsinized 1 × 10^6^ cells/mL with or without treatments and lysed them in lysis buffer 6 provided by the Proteome Profiler™ Array (Cat Nr: ARY003B R&D Systems, Minneapolis, MN, USA) following the experimental protocol mentioned in the manufacturer’s instructions. Briefly, 2 nitrocellulose membranes were used per sample in an 8-well multi-dish for a total of 4 samples. The samples were then treated with blocking buffer, washing buffer, and antibody cocktails. Membranes were exposed to Chemi Reagent Mix to detect the proteins.

### 4.9. Immunofluorescence Assay

Cells were grown on chamber slides (Nunc Lab Tek, ThermoFisher Scientific, Waltham, MA, USA) until confluency. They were then starved (RPMI1640 without FCS) for at least 48 h prior to the experiment followed. The cells were treated with or without anisomycin/DMSO. Briefly, the cells were washed with PBS, fixed with 4% paraformaldehyde for 30 min, and followed by blocking in immunofluorescence buffer (0.2% TritonX-100, 0.2% BSA, 0.2% Caesin, 5% goat serum DAKO, 0.2% gelatin, and 0.02% sodium azide) for 30 min. Beta-catenin primary antibody (1:1000, BD) was diluted in blocking buffer and incubated for 1 h. The cells were washed again in PBS solution and incubated with secondary antibody (Alexa 488, 1:2000, CST, ThermoFisher Scientific, Waltham, MA, USA) and Hoechst dye (1:1000, BD) diluted in blocking buffer for 30 min. Then, the slides were washed in PBS and closed with a drop of PBS:Glycerin (1:1) with a cover slip; they were imaged immediately.

For SiR-actin staining, 5 × 10^5^ cells were grown on chamber slides for 24 h followed by 48 h of starving as mentioned above. The cells were supplemented with melanoma complete medium for 24 h prior to treatments with anisomycin/DMSO. Cell fixation was performed as described above, followed by the addition of SiR-actin dye (1:500, Spirochrome) diluted in complete medium and incubation for 3 h. The cells were briefly washed with PBS and incubated with medium containing Hoechst dye (1:1000, BD) for 30 min. Then, the slides were washed in PBS and closed with a drop of PBS:Glycerin (1:1) with a cover slip. They were imaged immediately.

For the measurement of actin intensity, this code on Github was followed: imaging/measuring-cell-fluorescence-using-imagej.rst. Briefly, the image was opened in ImageJ software and converted to an 8-bit image. A boundary of the cell was drawn, and the area-integrated and mean gray value were measured. To subtract the background, an area without any cell was selected and the same process was repeated. The CTFC (corrected total cell fluorescence) was calculated as CTFC = Integrated Density − (Area of selected cell ∗ mean fluorescence intensity of background). The average CTFC was calculated for each block of slide with images in at least 3 areas. Experiments were performed in triplicate.

### 4.10. Immunoblotting

Cells were lysed with a radioimmunoprecipitation assay (RIPA) buffer (150 mM of NaCl, 15 mM of MgCl_2_, 1 mM of EDTA, 50 mM of HEPES, 10% glycerol, 1% Triton X-100, 1 tablet/mL each of phosphatase inhibitor (Roche, Basel, Switzerland) and protease inhibitor (Roche, Basel, Switzerland)) on ice for 30 min, and 20 µg of protein was analyzed using a standard Western blotting analysis. Protein quantification was conducted using a standard Bradford (BioRad, Hercules, CA, USA) assay. Cell lysates were collected 30 min or 24 h after 0.1 µM of anisomycin treatment or 10 µM of hydroxychloroquine treatment. The list of primary antibodies with their respective dilution and manufacturer are listed in Table 1 below. Anti-HSP90 (Cell Signaling, Danvers, MA, USA) was used as the loading control at a 1:2000 dilution. At least 3 antibodies were probed on the same membrane after cutting the membrane for various sizes by staining it with Ponceau red after the transfer protocol. All membranes were probed for 60 min at room temperature with secondary anti-rabbit or anti-mouse antibody (Cell Signaling, Danvers, MA, USA) at a 1:2000 dilution. The visualization was performed using an ECL chemifluorescent reagent (Invitrogen, Waltham, MA, USA) or ECL western bright Sirius/Quantum (Advantas, Farmington, NM, USA). For all immunoblotting experiments, we treated the p38–160915/130429 cells for only 30 min as longer incubation times with anisomycin resulted in unviable cells with limited amount of cell lysate. Experiments were repeated at least 3 times.

### 4.11. Statistical Analysis and Blinding Approach

The results are presented as the mean ± standard deviation or mean ± standard error representation of three independent experiments. A 2-way ANOVA test was used to measure categorical data, in the motility and spheroid assays. A *p*-value of less than 0.05 was considered statistically significant.

A partial blinding approach was followed for the analysis of RNA sequencing.

## Figures and Tables

**Figure 1 cancers-15-00877-f001:**
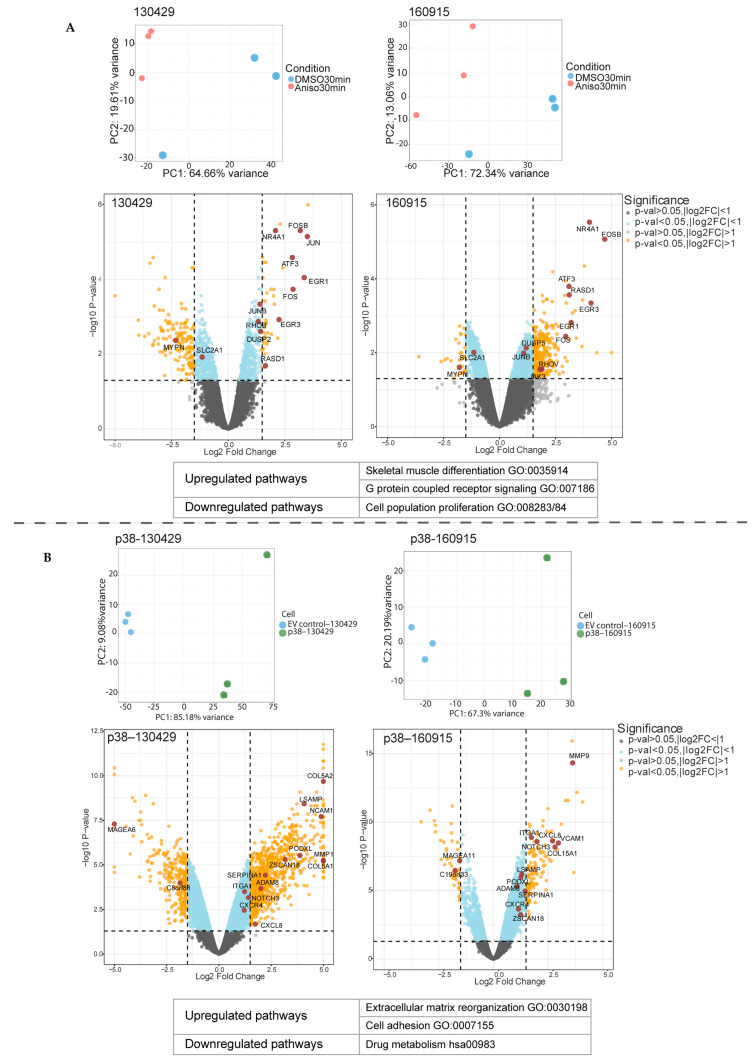
Distinct set of genes and pathways upregulated in the anisomycin-treated 130429/160915 cells and p38–130429/160915 cells. (**A**): Principal component analysis of 30 min anisomycin-treated 130429/160915 cells compared with DMSO-treated 130429/160915 cells (normalized + log2). Volcano plot showing differentially expressed genes (DEGs) in 30 min anisomycin-treated 130429/160915 cells compared with DMSO-treated 130429/160915 cells (FDR threshold: 0.05). The most upregulated and downregulated pathway as indicated by the over-representation analysis (ORA), *p*-value < 0.05. (**B**): Principal component analysis of p38–130429/160915 cells compared with EV control–130429/160915 cells (normalized + log2). Volcano plot showing differentially expressed genes (DEGs) in p38–130429/160915 cells compared with EV control–130429/160915 cells (FDR threshold: 0.05). Most upregulated pathway as indicated by over-representation analysis (ORA) and most downregulated pathway as indicated by KEGG, *p*-value < 0.05.

**Figure 2 cancers-15-00877-f002:**
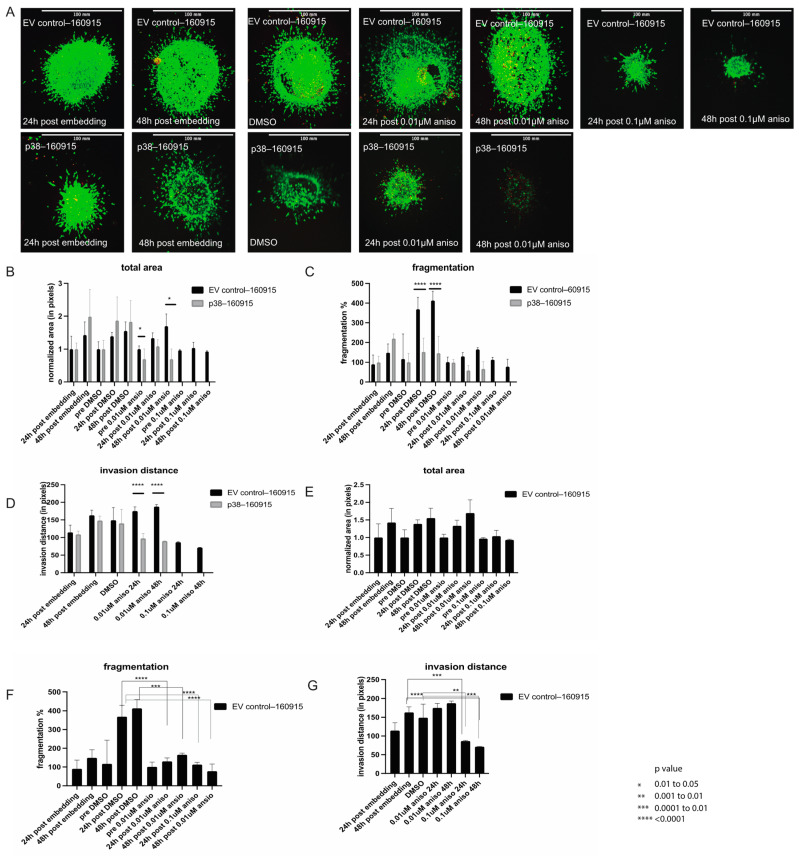
Invasion assay in spheroids and protein expression of MITF. (**A**): EV control–160915 and p38–160915 spheroids were stained with calcein and ethidium and compared at 24, 48 and 72 h post-embedding in collagen with and without anisomycin/DMSO. (**B**): Total area was compared at 24, 48 and 72 h post-embedding in collagen in addition to comparing the total area between DMSO, 0.01 µM, and 0.1 µM anisomycin treatments lasting 24 and 48 h in both EV control–160915 and p38–160915. A significant difference was observed between EV control–160915 and p38–160915 at 24 h post embedding in collagen and prior to anisomycin treatment (*p* = 0.0138). as well as 72 h post-embedding in collagen and 48 h post 0.01 µM anisomycin treatment (*p* = 0.0197). Bars labelled as pre-DMSO/aniso refer to the spheres before the respective treatments. (**C**): Fragmentation was compared at 24, 48, and 72 h post-embedding in collagen in addition to a comparison of the fragmentation between DMSO, 0.01 µM, and 0.1 µM anisomycin treatments lasting 24 and 48 h in both EV control–160915 and p38–160915. A significant difference was observed between EV control–160915 and p38–160915 24 and 48 h post-DMSO treatment (*p* < 0.0001). Bars labelled as pre-DMSO/aniso refer to the spheres before the respective treatments. (**D**): Invasion distance was compared at 24, 48, and 72 h post-embedding in collagen in addition to a comparison of the invasion distance between DMSO, 0.01 µM, and 0.1 µM anisomycin treatments lasting 24 and 48 h in both EV control–160915 and p38–160915. A significant difference was observed between EV control–160915 and p38–160915 24 and 48 h post-0.01 µM anisomycin treatment (*p* < 0.0001). Invasion distance was calculated using images from the confocal microscope. (**E**): Total area was calculated using the same methods as in B except that the comparison group is limited to EV control–160915 only. (**F**): Fragmentation was calculated using the same methods as in C except that the comparison group is limited to EV control–160915 only. A significant difference was observed between 24 h DMSO-treated spheroids and 0.01 µM anisomycin treated spheres at 24 and 48 h (*p* < 0.0001). (**G**): Invasion distance was calculated using the same methods as in D except that the comparison group is limited to EV control–160915 only. A significant difference was observed between 48 h post-embedding in collagen and 0.1 µM anisomycin treatment at 24 (*p* < 0.002) and 48 h (*p* < 0.0001). A significant difference was observed between DMSO-treated spheroids and 0.1 µM anisomycin-treated spheroids at 24 (*p* < 0.0027) and 48 h (*p* < 0.001). This is a representative picture of one replicate out of the three biological replicates. The Experiment was repeated at least three times with three biological replicates.

**Figure 3 cancers-15-00877-f003:**
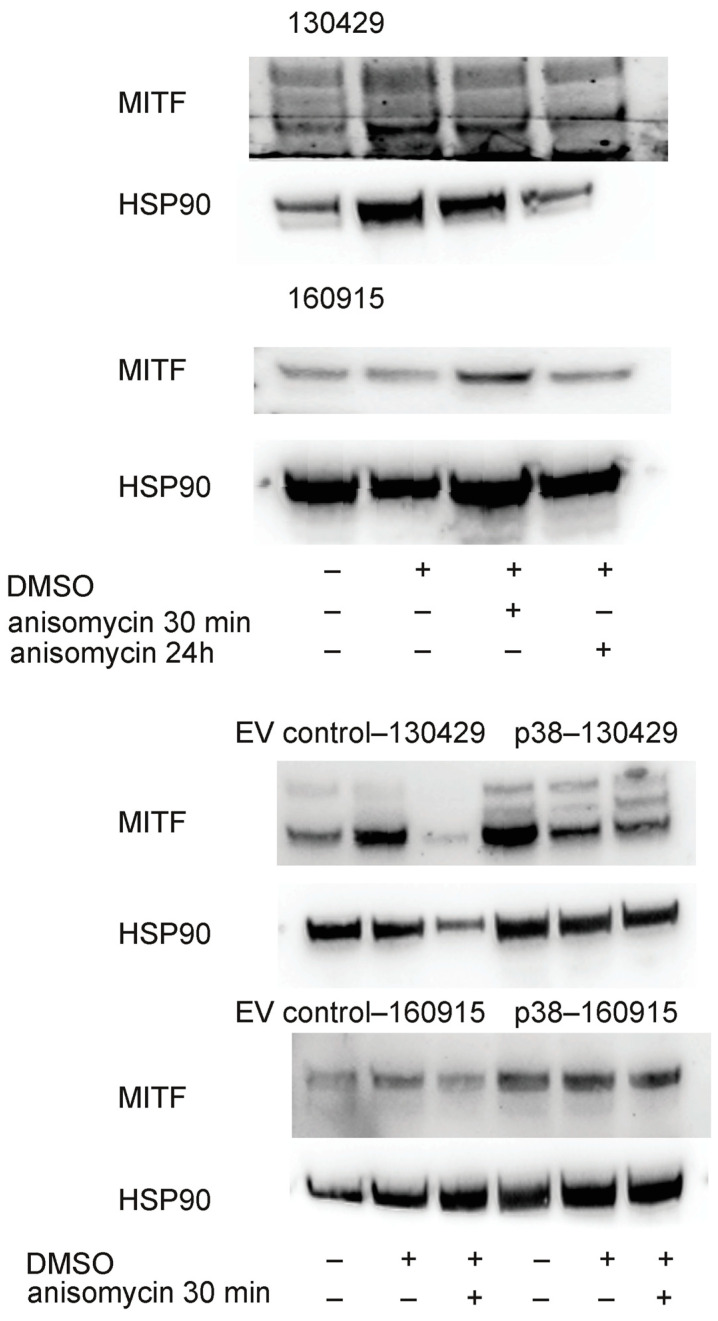
First and second panel: Protein expression of MITF with HSP90 as loading control under the untreated, DMSO, 0.1 µM anisomycin for 30 min, and 0.1 µM anisomycin for 24 h treatment conditions in cell lines 134029 and 160915. Third and fourth panel: Protein expression of MITF with HSP90 as loading control under the untreated, DMSO, and 0.1 µM anisomycin for 30 min treatment conditions in cell lines EV control–130429/160915 and p38–130429/160915.

**Figure 4 cancers-15-00877-f004:**
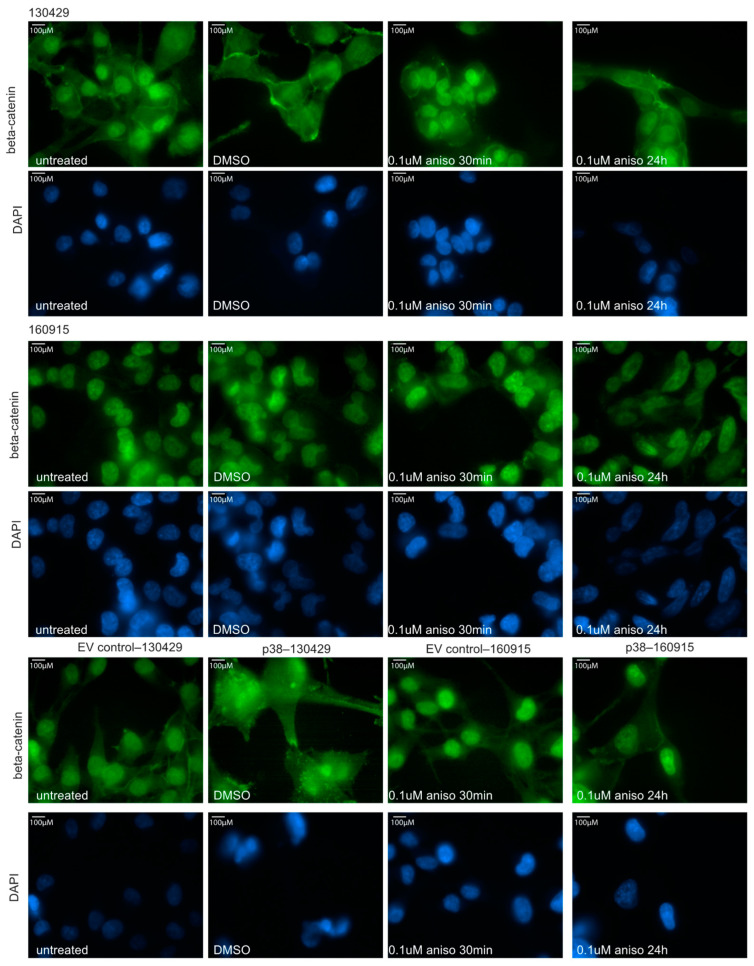
Beta-catenin expression and cell motility assay. Immunofluorescence staining of beta-catenin (green) and nucleus (blue) in cell lines 130429, 160915, EV control–130429/160915, and p38–130429/160915 under DMSO, 0.1 µM anisomycin for 30 min, and 0.1 µM anisomycin for 24 h.

**Figure 5 cancers-15-00877-f005:**
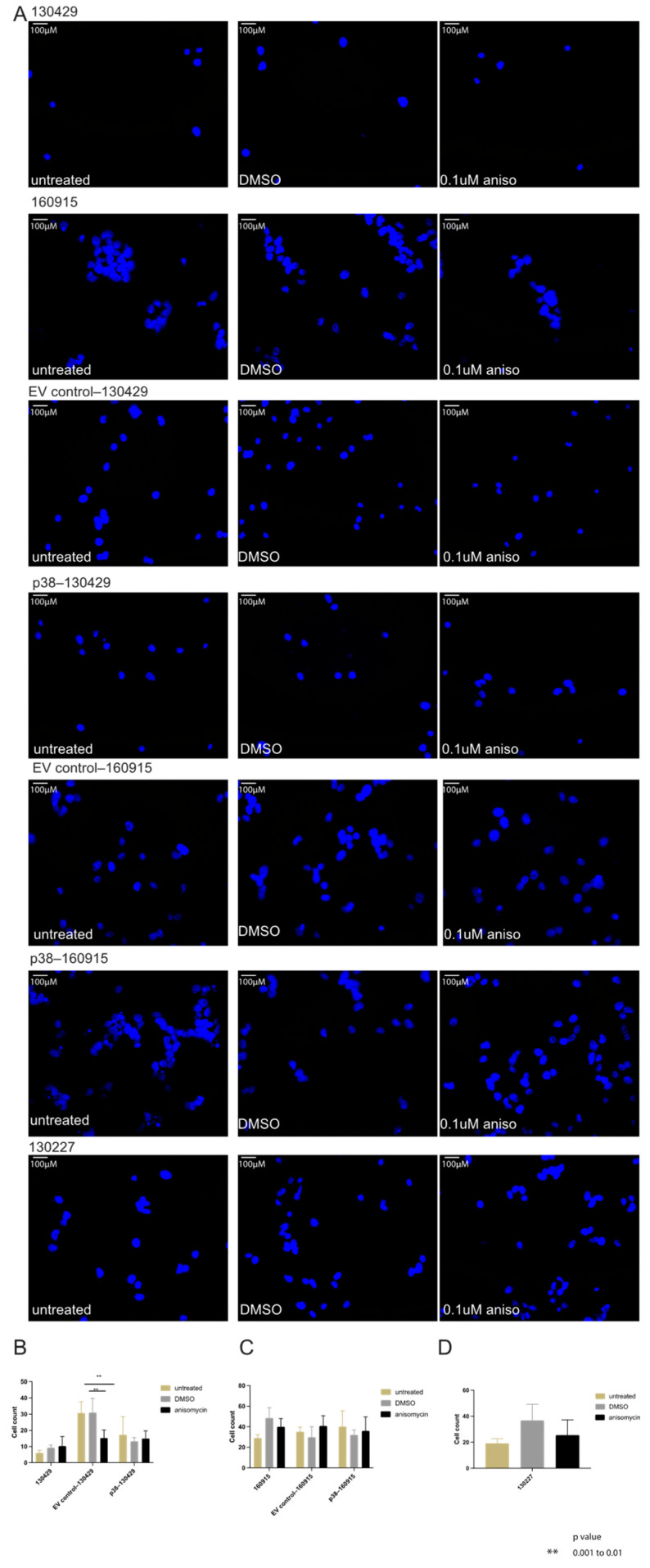
(**A**): Immunofluorescence staining of nucleus (blue) in cell lines 130429, 160915, 130227, EV control–130429/160915, and p38–130429/160915 under DMSO and 0.1 µM anisomycin for 24 h. (**B**): Cell counts from immunofluorescence staining of nucleus in cell type 130429, EV control–130429, and p38–130429 under varying treatment conditions. Significant difference observed in anisomycin-treated cells compared with DMSO-treated (*p* = 0.0034) and untreated cells (*p* = 0.004). (**C**): Cell counts from immunofluorescence staining of nucleus in 160915, EV control–160915, and p38–160915 under varying treatment conditions. (**D**): Cell counts from immunofluorescence staining of the nucleus in cell line 130227 under varying treatment conditions.

**Figure 6 cancers-15-00877-f006:**
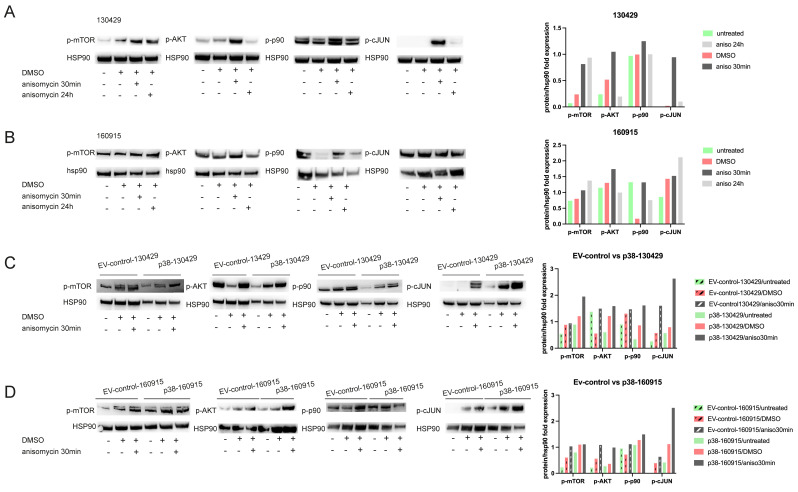
Transient phosphorylation of mTOR/AKT and cJUN pathway with short-term anisomycin treatment and sustained phosphorylation of mTOR in long-term anisomycin-treated and p38–130429/160915 cells. (**A**): Protein expression of phospho-mTOR, AKT, p90, and cJUN with HSP90 as the loading control under untreated, DMSO, and 0.1 µM anisomycin treatment for 30 min and 24 h in cell line 130429. Densitometry plot represents the expression of proteins as the fold change in relation to HSP90 expression. (**B**): Protein expression of phospho-mTOR, AKT, p90, and cJUN with HSP90 as the loading control under untreated, DMSO, and 0.1 µM anisomycin treatment for 30 min and 24 h in cell line 160915. Densitometry plot represents the expression of proteins as the fold change in relation to HSP90 expression. (**C**): Protein expression of hosphor-mTOR, AKT, p90, and cJUN with HSP90 as the loading control under untreated, DMSO, and 0.1 µM anisomycin treatment for 30 min in EV control–130429 and p38–130429 cell lines. Densitometry plot represents the expression of proteins as the fold change in relation to HSP90 expression. (**D**): Protein expression of phospho-mTOR, AKT, p90, and cJUN with HSP90 as the loading control under untreated, DMSO, and 0.1 µM anisomycin treatment for 30 min in EV control–160915 and p38–160915 cell lines. Densitometry plot represents the expression of proteins as the fold change in relation to HSP90 expression.

**Figure 7 cancers-15-00877-f007:**
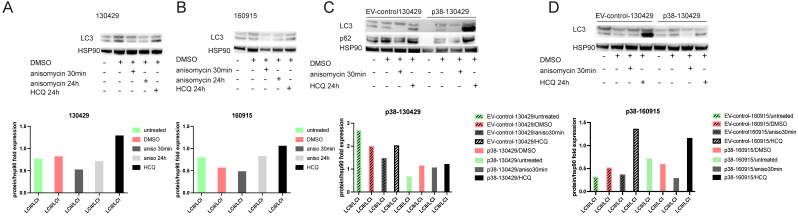
Transient suppression of autophagy by reduced conversion of LC3 with p38 activation. (**A**): Protein expression of LC3 with HSP90 as the loading control under untreated, DMSO, 0.1 µM anisomycin treatment for 30 min or 24 h, and 10 µM HCQ treatment for 24 h in cell line 130429. Densitometry plot represents the expression of proteins as the fold change in relation to HSP90 expression. (**B**): Protein expression of LC3 with HSP90 as the loading control under untreated, DMSO, 0.1 µM anisomycin treatment for 30 min or 24 h, and 10 µM HCQ treatment for 24 h in cell line 160915. Densitometry plot represents the expression of proteins as the fold change in relation to HSP90 expression. (**C**): Protein expression of p62 and LC3 with HSP90 as the loading control under untreated, DMSO, 0.1 µM anisomycin treatment or 24 h, and 10 µM HCQ treatment for 24 h in EV control–130429 and p38–130429 cell lines. Densitometry plot represents the expression of proteins as the fold change in relation to HSP90 expression. (**D**): Protein expression of LC3 with HSP90 as the loading control under untreated, DMSO, 0.1 µM anisomycin treatment for 30 min or 24 h, and 10 µM HCQ treatment for 24 h in EV control–160915 and p38–160915 cell lines. Densitometry plot represents the expression of proteins as the fold change in relation to HSP90 expression.

**Figure 8 cancers-15-00877-f008:**
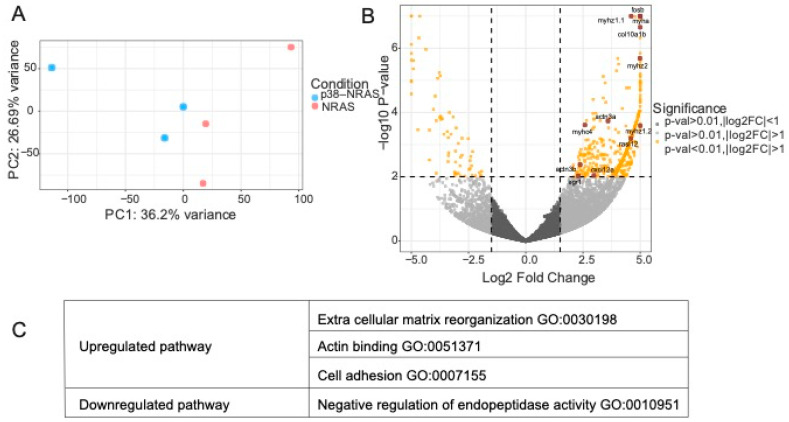
RNA sequencing analysis of p38–NRAS^Q61K^-and NRAS^Q61K^-derived zebrafish tumor tissues. (**A**): Principal component analysis of p38–NRAS^Q61K^ tumors compared with NRAS^Q61K^ tumors (normalized + log2). (**B**): Volcano plot showing differentially expressed genes (DEGs) in p38–NRAS^Q61K^ tumors compared with NRAS^Q61K^ tumors (raw threshold; 0.01). Most upregulated and downregulated pathway by over-representation analysis (ORA, *p*-value < 0.01). (**C**): Most upregulated and downregulated pathways in tumors derived from p38– NRAS^Q61K^ compared to NRAS^Q61K^ as represented by ORA, *p*-value < 0.05.

**Table 1 cancers-15-00877-t001:** List of primary antibodies.

Primary Antibody	Dilution	Manufacturer	Primary Antibody	Dilution	Manufacturer
Phospho cJUN 2361	1:1000	Cell Signaling	Phospho-p90 9341	1:1000	Cell Signaling
Total cJUN 9165	1:1000	Cell Signaling	Total p90 9355	1:1000	Cell Signaling
Phospho-mTORS2481 2974	1:1000	Cell Signaling	Phospho-pRAS40 13175	1:1000	Cell Signaling
Phospho-mTORS2448 5536	1:1000	Cell Signaling	Total pRAS40 2610	1:1000	Cell Signaling
Total mTOR 2983	1:1000	Cell Signaling	Phospho-PDK1 3061	1:1000	Cell Signaling
Phospho-AKTS437 4060	1:1000	Cell Signaling	Total PDK1 3062	1:1000	Cell Signaling
Phospho-AKTT803 13038	1:1000	Cell Signaling	HSP90 4877	1:2000	Cell Signaling
Pan AKT 4691	1:2000	Cell Signaling	P62 sc28359	1:250	Santa Cruz
Phospho-p70 9234	1:1000	Cell Signaling	BECLIN 3738	1:1000	Cell Signaling
Total p70 9202	1:1000	Cell Signaling	ATG4 5299	1:1000	Cell Signaling
Phospho GSK-ß 9322	1:1000	Cell Signaling	LC3 ab51520	1:3000	Abcam
Total GSK-ß 9315	1:1000	Cell Signaling	MITF ab12039	1:1000	Abcam

## Data Availability

Data is available upon request to corresponding authors.

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
