# Peer review of "P38 Mediates Tumor Suppression through Reduced Autophagy and Actin Cytoskeleton Changes in NRAS-Mutant Melanoma"

_cancers, 2023, doi:10.3390/cancers15030877_

Round 1

Reviewer 1 Report

The manuscript “P38 mediates tumor suppression through reduced autophagy and actin cytoskeleton changes in NRAS mutant melanoma” (by Banik et al.) reveals differences of the function of p38 kinase in BRAF mutated and NRAS mutated melanomas. This is quite an interesting finding - p38 behaved rather as tumor suppressor in NRASmut melanomas (unlike in BRAFmut melanomas).

Major and minor concerns:

Abstract: 

line 29-30: ”….here that p38 specifically plays the role of a tumor suppressor in NRAS mutant melanoma at least partially through the mechanism of mTOR upregulation, suppressed autophagy”… How the role of p38 as a tumor suppressor is reconciled with its activity to activate mTOR pathway and suppression of autophagy? Please explain.

Minor: Introduction: lines 38-39: HRAS and KRAS play only marginal role in melanoma compared to NRAS. Consider if it necessary to mention these genes.

Minor: Line 43- …”Given that one-third of melanoma patients have NRAS”…  Most papers show that the average mutational frequence of NRAS in melanoma is only about 15%.

Results:

Some other reports do not fully support your results (Puujalka et al., cited). Do you have some explanation?

“In summary, our results indicate a transient autophagy deficiency upon anisomycin treatment. p38 activation results in transient activation of phospho-p90 and AKT and stable activation of phospho-mTOR.”

There remain some feeling of confusion, eg. short and long effects of anisomycin treatment, and other.

It would be really beneficial to have a positive control compared with p38, eg. some typical prooncogenic MAPK kinase from the two remaining subtypes of MAPK pathway, tested in NRASmut melanomas. It would clearly show the exception of p38 signaling from other prooncogenic event in similar pathway.

Some results are described in very great details that could attenuate the attention from the main story.

Do you have some (at least speculative) molecular explanation why p38 may behave differently then other classical MAPK kinases in NRASmut melanomas?

Discussion:  

Lines 472-3: ..”mechanism of action of p38 induced tumor suppressive effects in NRAS mutant melanoma is through the activation of mTOR pathway which in turn leads to suppressed autophagy.”

This might be exceptionally possible, but increased mTOR pathway and lowered autophagy is generally considered as pro-oncogenic mechanisms.

Author Response

Dear Reviewer,

We thank you for your useful suggestions and comments. We have answered your questions in the attached document as well as updated the manuscript and figures. Thank you for your time.

Reviewer 2 Report

The authors studied p68 in NRA mutant melanoma and identified reduction of auto-Nagy and actin cytoskeleton changes.

A set of describing results and performing new analysis is needed.

1. The authors should describe better their RNAseq methodology in giving for each package their versions and the R version that they used in their analysis and the parameters. Moreover, they should provide in the Fig. 1 and 6 the KEGG numbers and GO numbers on top of the name of pathway to allow to have the list of genes and give the q-value and log fold or OR. They need to provide also this information the result paragraph.

2. The authors studied first after RNAseq invasion. They should first study the proliferation gene set associated with melanoma (https://doi.org/10.1158/0008-5472.CAN-07-2491 and http://www.jurmo.ch/work_model.php) as their RNAseq shows a decrease in proliferation genes. Maybe more positive findings will be identified. They could perform a heatmap with the invasion and proliferation genes for their different datasets. on top of volcano plot.

3. in the Fig. 2., the author does not provide any image and data at time 0 and so it is very difficult to fully analysis the data. Moreover, the authors need  to replace all their barplot by box plot plus dot to visualise the different replicate data and they need to provide the number of replicates they performed and put the pvalue instead of * (https://www.nature.com/articles/nmeth.2807 , https://onlinelibrary.wiley.com/doi/full/10.1016/j.pmrj.2016.02.001 you can still do it for 3 datapoints, but statistically we prefer 5 replicates to show any distribution)

4. the author should show the proliferation gene or invasion genes that most altered by 30min anusomycin treatment from their data on top of their MITF analysis. Maybe MITF was not the best representative for their datasets.

5. the authors need to provide the catalogs number of their different reagents in each assay, without not forgetting for different antibodies for WB and IF they performed

Author Response

Dear Reviewer,

Thank you for your useful suggestions and comments. We have answered your questions in the attached document as well as updated the manuscript and figures. We thank you for your time.

Reviewer 3 Report

This is a well written manuscript on the switch of authophagic phenotype depending on whether BRAF- or NRAS- driven melanoma cells are investigated.  The kinase p38 is described as the central switch to reduce authophagy in NRAS mutated melanoma cells via a sustained mTOR phosphorylation. However, cell motitility and invasion were not affect by p38 activation.

The conclusions are sound and supported by solid experimental data.

Major Issues:

#1 Some of the figures are overloaded with panels, so that it is very difficult or even impossible to read axis labels, columne treatments or statistics.

Hence, rearrangement of Figs. 2, 3, 4 and 5 is highly recommended or separation into new figures.

#2 Magnification of microscopic images is not stated nor indicated by a bar in the Figs. 2 and 3.

Author Response

(The authors gave the same response as above.)

Round 2

Reviewer 2 Report

The Authors mentioned that they have already studied the proliferation with over expression of p38. However, if they claimed it, several times the authors said that they previous demonstrated without providing the paper ( line 86-87) or line 140-142. It is important that the authors provided the reference for such claims.

The authors reformatted their figures, but it seems that they keep Fig 3A (line 230), but it should be Figure 4.

The authors did not fully answer the questions.

1) it is still difficult to understand how the RNAseq analysis driven the following analysis as they did not use their RNAseq findings to decide the follow-up analysis. Moreover, they studied the invasion phenotypes, that is a switch phenotype with proliferation. The authors need to improve the transition between RNAseq data and the invasion study.

2) the authors need to add in the legend and results the number of replications and for the figures, they need to indicate in the legend that they show only one representative of their x replicates to inform readers that the number of replicates they performed it.

3) the authors did not answer about the need to provide the catalog numbers of their reagents, kit, and antibodies to help readers to try to replicate their findings with the same or close elements. For example, We can have multiple antibodies for the same proteins from the same company, but with not the same quality. It is important to provide them for the readers.

Author Response

Dear Reviewer,

Thank you so much for your insightful comments. Please find attached a detailed response to your report and a revised manuscript.

Thank you.

Round 3

Reviewer 2 Report

the authors answered the different comments